# Opportunities and challenges of a novel cardiac output response to stress (CORS) test to enhance diagnosis of heart failure in primary care: qualitative study

Sarah Charman,[1,2] Nduka Okwose,[1,2] Gregory Maniatopoulos,[3,4] Sara Graziadio,[4] Tamara Metzler,[1] Helen Banks,[1] Luke Vale,[3] Guy A MacGowan,[1,2] Petar M Seferović,[5] Ahmet Fuat,[6] Christi Deaton,[7] Jonathan Mant,[7] Richard F D Hobbs,[8] Djordje G Jakovljevic[1,2,9]

For numbered affiliations see end of article.

**Correspondence to**
Dr Djordje G Jakovljevic;
djordje.Jakovljevic@newcastle.ac.uk

## ABSTRACT

**Objective** To explore the role of the novel cardiac output response to stress (CORS), test in the current diagnostic pathway for heart failure and the opportunities and challenges to potential implementation in primary care.

**Design** Qualitative study using semistructured in-depth interviews which were audio recorded and transcribed verbatim. Data from the interviews were analysed thematically using an inductive approach.

**Setting** Newcastle upon Tyne, UK.

**Participants** Fourteen healthcare professionals (six males, eight females) from primary (general practitioners (GPs), nurses, healthcare assistant, practice managers) and secondary care (consultant cardiologists).

**Results** Four themes relating to opportunities and challenges surrounding the implementation of the new diagnostic technology were identified. These reflected that the adoption of CORS test would be an advantage to primary care but the test had barriers to implementation which include: establishment of clinical utility, suitability for immobile patients and cost implication to GP practices.

**Conclusion** The development of a simple non-invasive clinical test to accelerate the diagnosis of heart failure in primary care maybe helpful to reduce unnecessary referrals to secondary care. The CORS test has the potential to serve this purpose; however, factors such as cost effectiveness, diagnostic accuracy and seamless implementation in primary care have to be fully explored.

## INTRODUCTION

Heart failure (HF) is a complex clinical syndrome resulting from impaired heart function.[1 2] It is recognised globally as a challenging public health burden and is characterised by debilitating symptoms for the patient, including breathlessness, oedema and fatigue and often coexists with other comorbidities.[3] Signs and symptoms are non-specific and therefore difficult for primary care physicians to produce an accurate diagnosis and referral.[4] Increasing ageing population

### Strengths and limitations of this study

► We have recently developed a novel cardiac output response to stress (CORS) test which has potential to improve diagnostic accuracy of heart failure in primary care.
► The interviews with 14 primary and secondary healthcare professionals allowed exploration of the opportunities and challenges for the implementation of the CORS test.
► Study participants were recruited from one geographic region, that is, the North of England.
► No patients were interviewed.

and improved outcomes in acute coronary syndrome have led to a growing incidence and prevalence of HF.

Early diagnosis of HF is crucial to improve survival, reduce hospitalisation and costs.[1 5] Primary care represents the first point of care in the clinical care pathway (figure 1) for patients presenting with symptoms of HF and general practitioners (GPs) play a crucial role in early identification[6 7] of patients. Evaluation of patients begin with a detailed medical history and performance of diagnostics tests. The diagnostic tests currently available to GPs include ECG[8]), and serum natriuretic peptides (N-terminal prohormone of brain natriuretic peptide [NT-proBNP]) tests. ECGs are insufficiently sensitive to detect HF and NT-proBNP test is only accurate as a test to exclude HF diagnosis; however, the later test can lead to a significant number of false positives due to its low specificity.[9–11] This has resulted in an unnecessary increase in referrals of patients to secondary care as >65% of suspected patients with HF referred from primary to secondary care do not have diagnosis confirmed following echocardiography

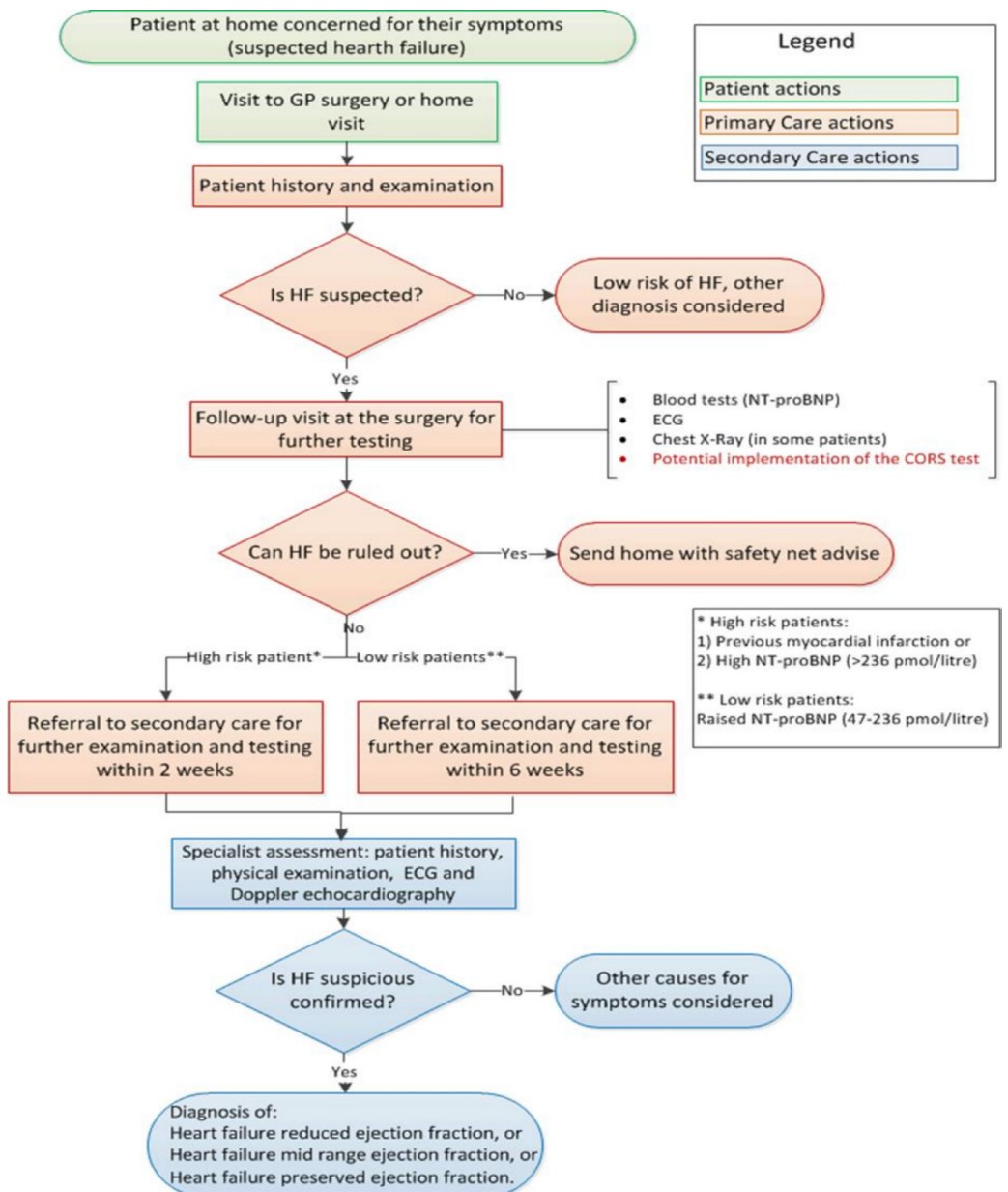

**Figure 1** Current clinical pathway for suspected heart failure (HF) diagnosis described by healthcare professionals in this study and adopted from the guidelines from the National Institute for Health and Care Excellence for Chronic Heart Failure.[2 3] CORS, cardiac output response to stress; GP, general practitioner; NT-proBNP, N-terminal prohormone of brain natriuretic peptide.

and specialist review.[4 8 12 13] This creates undue worry for the patients as there is a delay in care of patients with the average waiting time for investigation being 67 days.[14] This also poses unnecessary financial and human resource burden to secondary care providers as time and resources could have been used more effectively for patients who need the attention of secondary care specialists.[1 2 15]

We have recently developed and confirmed acceptable reproducibility of a novel, easy-to-use, non-invasive cardiac output response to stress (CORS) test (figure 2) for the evaluation of cardiac function in primary care.[16] The CORS test has the potential to be used by healthcare professionals to improve diagnostic accuracy for suspected HF and provide an objective measure of

cardiac function as a 'rule in' test. Prior to conducting a large trial on further effectiveness of the CORS test, it was important to understand how primary and secondary healthcare professionals perceive this new diagnostic test. Information from healthcare providers would help us understand if there was a relevance of the CORS test and also help us make better judgement on the design of a larger trial if we thought this was plausible. The objectives of this study were first to explore the role of the CORS test in the current clinical care pathway for suspected HF, and second to identify the potential opportunities and challenges to potential implementation in care of patients.

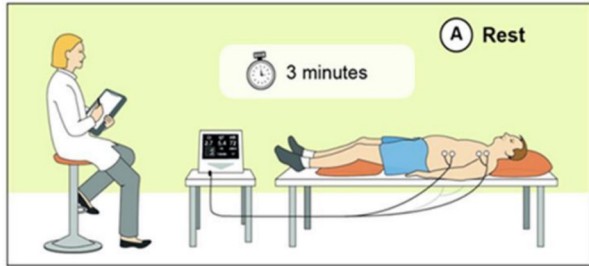

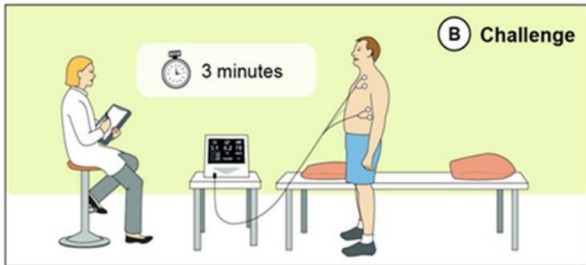

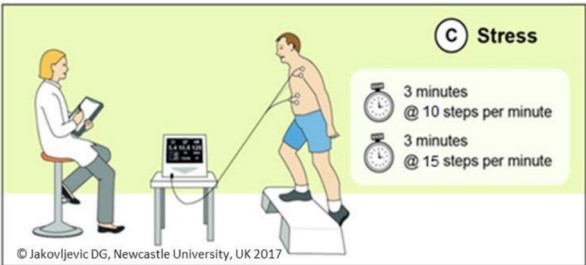

© Jakovljevic DG, Newcastle University, UK 2017

**Figure 2** Cardiac output response to stress test. The cardiac output response to stress test consists of three phases: rest, challenge and stress exercise. Each phase lasts for 3 min, and stress phase integrates additional 3 min to increase intensity and metabolic demand (from 10 to 15 steps/min).

## METHODS
### Study design, setting and participants
Semistructured in-depth interviews (14 in total; table 1) were conducted between December 2017 and April 2018 in the North of England. Participants were purposely selected according to their role and involvement in the management of patients with suspected or diagnosed HF in both primary and secondary care. Potential interviewees were sent an email invitation which briefly outlined the aims of the study. Those agreeing to participate were invited to recommend additional candidates for interview. Data were collected at the workplace of study participants. All interviews were digitally recorded, anonymised and transcribed in full.

### One research associate (SC) conducted the fieldwork
Interviews were typically around 60 min in length and conducted on an individual, face-to-face basis. The final sample is described in table 1. First, participants were asked questions on the current clinical care pathway for HF. Then, a video demonstration of a patient completing the CORS test was shown with an explanation of each phase of the test provided by the interviewer (duration: 02:41 min). Finally, questions were asked about the CORS test and potential barriers and facilitators to its implementation. Examples of questions asked which relate to the main objectives of the study include: (1) Does the current clinical care pathway lead to you seeing many patients who are unlikely to receive a HF diagnosis in secondary care? (2) Do the referral time from suspected HF diagnosis in primary care to confirmed diagnosis need to be improved? (3) What are your thoughts about this potential improvement in diagnosis being implemented in the clinical care pathway for HF? (4) Where in the clinical care pathway for suspected HF diagnosis would you expect this test to be performed? (5) Do you anticipate any immediate barriers to implementing this test as part of your routine clinical care for a patient with a suspected HF diagnosis in your practice? Prompts were used generate a deeper understanding of participants' views about the questions asked (see online supplementary appendix for full list of interview questions and prompts).

### The CORS test
The CORS test (figure 2) has been previously described[16] and consists of four phases: (1) rest, where the patient lies in the supine position for 3 min; (2) challenge, where the patient remains in the standing position for 3 min;

| Table 1 | List of interviewees | | | | |
|---|---|---|---|---|---|
| Organisation | No of interviews | Interviewee | Sex | Years in practice | Practice list size |
| Practice 1 | 2 | GP<br>Practice manager | M<br>M | 40<br>2 | 5512 |
| Practice 2 | 3 | GP<br>Practice nurse<br>Practice manager | M<br>F<br>F | 15<br>15<br>2 | 5140 |
| Practice 3 | 1 | GP | F | 26 | 6491 |
| Practice 4 | 2 | GP<br>Healthcare assistant | M<br>F | 25<br>13 | 11091 |
| Community setting | 3 | Heart failure specialist nurse | F | 13, 5, 17 | |
| Hospital site | 3 | Consultant cardiologist | F, Mx2 | 33, 30, 16 | |
| Total | 14 | | | | |

GP, general practitioner.

(3) and (4) stress exercise, where a patient completes a step test with continuous haemodynamic measurements (ie, cardiac output and cardiac index) using non-invasive, electrical signal processing technology.

## Patient and public involvement

Patients/public representatives were not directly involved in the present qualitative study. However, their views about the design of the main study, which aims to assess the clinical and cost effectiveness of the CORS test to improve diagnosis of HF in primary care, have been explored.

## Data analysis

Given the exploratory nature of the present study, an inductive approach to data analysis was adopted. Thematic analysis was used because it is a flexible method that allowed themes to emerge freely from the data obtained.[17] All interviews were transcribed verbatim. The following analysis procedure was undertaken: (1) two researchers independently read and reread interview transcripts (SC and NO) in order to become adequately familiarised with the data; (2) both researchers independently applied codes to segments of data in all interview transcripts to develop initial themes; (3) due to the large volume of data generated from the interviews, several initial codes were generated and were later cut down to a final set agreed by both researchers. The final codes were collated and this formed the basis from which themes were conceptualised. Supporting direct quotes from participants were subsequently selected for each theme. These were later discussed and approved by the research team.

## RESULTS

Analysis of the data generated four themes relating to opportunities and challenges surrounding the implementation of the new diagnostic technology: (1) novel diagnostic tests are required to improve HF diagnosis; (2) evidence required to improve the diagnostic accuracy of the CORS test; (3) patient exclusion and suitability/patient population for the CORS test and (4) financial implications of the CORS test within primary care.

### Theme 1: novel diagnostic tests are required to improve HF diagnosis

All GPs and consultants agreed that new diagnostic tests vital to improve HF diagnosis and were very positive about the potential role of the CORS test in primary care to aid diagnosis of HF and provided reasons regarding the large numbers of undiagnosed cases of the condition.

> Personally, I think this would be a great idea, because I think it would be, if it's more accurate, it would actually certainly reduce wastage in terms of referrals. And it would allow us to make the diagnosis much earlier and manage the patient based on that, really…. Well, you're talking to someone who believes that 70 per cent of HF is undiagnosed. And I don't know what the figure is, but most HF, I believe, is

currently diagnosed in the A&E (emergency admission unit)….So, you know, you're talking to someone who thinks that anything we can do which actually makes it easier to diagnose HF in primary care is a good thing. (GP, Practice 2).

Participants suggested that HF was difficult to diagnose with the patient often presenting with more than one complication or comorbidity, usually when disease exacerbates.

> If somebody came in who had had a previous MI and said, 'My ankles have swelled up. I can't lie flat in bed at night,' and they had nice crackles at the bottom of their lungs, you would probably be right there and then saying, 'I think this might be related to your heart,' and so on. But sometimes it's much less obvious. It's just somebody presenting with breathlessness and it could be anything from heart failure to lung cancer, COPD. I mean I'm thinking of a particular patient who did end up having, in the end, quite severe HF but it wasn't obvious at all. (GP, Practice 3).

It was also acknowledged by secondary care participants that in primary care HF is difficult to diagnose based on the limited diagnostics available in primary care including natriuretic peptides, which can be elevated due to increasing age or other comorbidities.

> I think you have to see quite a high proportion of people who do not have HF to avoid missing it. I also think it's quite a hard diagnosis in primary care because the blood tests are a very blunt tool. They're great for rule out but they do not rule in. Because the levels go up with age, they go up with multiple comorbidities. Many patients who are thought to have HF, actually will have a raised level, even if they turn out not to have HF. (Consultant Cardiologist 2, Hospital site).

One of the Consultant Cardiologists was particularly sympathetic with these concerns and acknowledged that the point of referral was different between practices and noted staff '*experience as well as knowledge and skill and confidence*' play an important role in this process.

Having a simple test like the CORS test could make identification of patients for referrals more effective, thus reducing the burden on secondary care providers. Consultants acknowledged that they often receive many inaccurate referrals which led to wasted consultant time.

> That's the issue. We're seeing too many patients who we don't really need to see. If they showed me an echo and ECG [report], I would say, 'I don't need to see them.' But the way it's set up, it's very inefficient. (Consultant Cardiologist 1, Hospital site).

Although, the potential value of the CORS test in primary care was clear, a variety of responses were given regarding the designated professional to deliver the CORS test, that is, GPs and nurses suggested either a practice

nurse or healthcare assistant. The healthcare assistant agreed that they could deliver this test given their experience of performing ECGs but would need practice and training. Practice managers suggested that nurses could potentially be the best person to deliver it.

For consultant cardiologists, a qualified nurse would be the best person to administer the test in primary care but others suggested local GPs practice should guide the selection of the most appropriate person for the delivery of the CORS test.

> you need somebody who can say, 'Keep going,' but you also need someone to say, 'No, stop. You look absolutely awful, stop.' (Consultant Cardiologist 2, Hospital site).

### Theme 2: evidence required to improve the diagnostic accuracy of the CORS test

Participants highlighted the role of scientific evidence in implementing a pathway for the new diagnostic technology.

> Yes. I would want to know what's the evidence for it, how clear cut is the interpretation and actually, if this is just going to be another pre-test that is done before the people go, then what exactly is the role of the clinic. (GP, Practice 3).

Although improvements in both referral accuracy and patient care were considered to be important for secondary care clinicians, it was difficult to determine where the CORS test would fit in the current clinical care pathway without knowing its efficacy. Secondary care clinicians reported that the implementation of the new diagnostic would be less useful in secondary care with the patient already receiving an echocardiogram.

> If it's proven to be effective in terms of being able to distinguish between possible HF or refuted diagnosis of HF, then I think it's useful. In terms of where it would fit into a pathway? Then it would appear to be that it would be at the start of the pathway before referring to secondary care. (Consultant Cardiologist 3, Hospital site).

Successful implementation of the CORS test would be dependent on the confirmation of clinical and cost effectiveness and convincing both GP partners and nurses regarding the clinical need for it and the value of completing the test and providing good patient care.

> You need more evidence, is the key bit. So it needs the evidence behind it. I think you might need assessment through NICE to see what they think it is as an additional test. (Consultant Cardiologist 2, Hospital site).

Based on the comments from themes 1 and 2, it was obvious that the CORS test was vital in the primary care setting. Figure 1 shows a flow chart that describes the current clinical care pathway for the diagnosis of HF as adapted from the National Institute for Health and Care Excellence (NICE) guidelines. The potential place of the CORS test has been inserted into the pathway to provide better context of the discussion.

### Theme 3: patient exclusion and suitability/patient population for the CORS test

A number of potential patient characteristics that could lead to patient exclusion were identified, for example, mobility, possible difficulties lying down, need of a balance aid, suggesting that not every patient would be suitable. Exercise was identified as potentially challenging. The number of patients who could potentially benefit from this technology could be increased if the duration of phases 3 and 4 could be reduced.

> We have some patients with established HF for example, you don't even recommend they weigh themselves every day because of the safety of stepping on to the scales. So I think there needs that recognition that some patients may not be suitable for it. (Consultant Cardiologist 2, Hospital site).

Other participants suggested possibility of using a set of parallel bars. GPs estimated a 50%–80% rate of uptake of the CORS test while the practice nurses were far more optimistic with an estimated 80% uptake.

### Theme 4: financial implications of the CORS test within primary care

A major concern raised about the implementation of the CORS test was on the logistics and overall cost implication of integrating the test into routine practice. Interviewees were sceptical about the implementation of the CORS test and suggested that they would need greater convincing about the time needed to administer the test, available space, cost of equipment and the required training expected to deliver the test efficiently.

An additional appointment would be required to implement the CORS in primary care. A healthcare assistant suggested an integration of the CORS test during the current ECG and biomarker testing appointment. The practice managers acknowledged that an additional appointment will incur an additional cost to the practice but also the availability of appointments would be a potential issue.

> Because, it wouldn't replace existing appointments [but require] additional appointments. That in itself adds a cost to the practice, that we would have to look at….That is the overall driver. You know, that assessment, obviously, has to be made at a partner and business level. The second thing, that always drives us, is cost. The third thing is impact on availability of appointments. Of course, it's availability of appointments is about the only thing as a practice we are actually measured on. (Practice Manager, Practice 1).

Practice managers identified the space for running the tests and storage of equipment as a potential issue.

Differences in the building capacity were acknowledged between practices by the practice managers. A practice nurse suggested that the receptionists would need to be aware what room the equipment was in and follow similar booking process as for the ECG. On the contrary, it was suggested that a potential setting for delivery of the CORS could be at the community setting such as community diagnostic centres or outreach diagnostic facility.

> There's increasingly, a feeling that not all practices can deliver all services at all times. However, it is more reasonable to work to the premise that all the patients of all the practices should have access to all the tests they need. But maybe just not in their own building. (GP, Practice 2).

The cost of the equipment was also acknowledged as a potential barrier to the implementation. One approach to commissioning a new pathway is to incentivise primary care through negotiation between the primary care contractors and their clinical commissioning groups.

> I wouldn't expect them to supply it. I would expect them to fund the equipment initially. They would normally pass on subsequent maintenance costs to us. I would also expect them to commission this as an enhanced service. So, that we can recoup the manpower and equipment costs of doing it. (Practice Manager, Practice 1).

Training was reported as a necessity by all healthcare professionals in primary care. It was suggested that the professional delivering the test would need to be trained and the professional interpreting the results. For example, a healthcare assistant could deliver the test but a GP would interpret it, in much the same process as currently occurs for ECGs.

> From a management perspective, my real concern is also how complicated the technology is. How easy it is for a practice nurse, who may not be using it day in day out to operate it?… You've done this test…How do you then get it to the doctor?. (Practice Manager, Practice 1).

## DISCUSSION

There has been an increase in the development and use of diagnostic tests in the last decade. This number will rise considerably taking into account the ageing population of developed countries and the number of people with chronic conditions.[18] There is a clinical need to improve point of care diagnostic tests available in primary care, in particular quick and high specificity tests to diagnose HF. This would improve treatment and reduce inaccurate referrals to secondary care thus improving overall efficiency and quality of care.[18] The present study captured primary and secondary care healthcare professional's perceptions for the implementation of a new diagnostic test. Our findings have identified the clinical role and potential value of the

CORS test and challenges that would need to be addressed during the next phase of development of the test.

Our findings suggest that research evidence on the clinical value of the CORS test is vital before implementation. The CORS test is the first of its kind to attempt to diagnose HF in primary care and there are no studies to compare with; however, we are currently assessing the sensitivity and specificity of the CORS test which will provide more information for the implementation process.

In addition, not every patient would be suitable for to CORS test due to limitation of movement. We plan to make the CORS test accessible for the majority of patients entering the clinical care pathway by shortening phases 3 and 4 and considering alternative positions (sitting or passive leg raise) using supporting aids during the exercise phase.

Similarly, the implementation of the CORS test requires adequate funding and support from clinical champions and commissioners. Although there is a general consensus that point of care tests, like the CORS test, increase 'turn-around-time' for results[19 20] and prevents additional visits to secondary care,[21] the financial implications of the CORS test within primary care, such as cost of tests, space, training and time need to be accounted for. These challenges which has been reported previously[22 23] may be addressed by positioning the CORS test within a community setting (ie, community diagnostic centres or outreach diagnostic facility). This would potentially reduce the burden on the practice to deliver the test.

### Strengths and limitations

This is the first study to explore the opportunities and challenges for the implementation of the CORS test. The perspective of primary and secondary healthcare professionals was considered, which ensured a more reflexive analysis of data.[24] This approach allowed the identification of suboptimal communication between primary and secondary care in the current HF clinical care pathway, which delays diagnosis and results in inaccurate referrals and wasted consultant time.

Given that participants were selected only from a region of the North of England and in small numbers meant that our findings may have limited generalisation.

Data were analysed thematically using an inductive approach, which led to the saturation of themes at the analysis stage.[25] However, this was the first iteration of the analysis of opportunities and challenges for the CORS test to inform further refinement of the CORS test.

### FUTURE DIRECTIONS

After collection of the necessary evidence, an application to the NICE's Diagnostic Assessment Programme will be attempted to increase the possibilities of widespread implementation and adoption in primary care.

**Author affiliations**
¹Cardiovascular Research Centre, Institutes of Cellular and Genetic Medicine, Faculty of Medical Sciences, Newcastle University, Newcastle upon Tyne, UK

[2]Cardiology Department, Freeman Hospital and Royal Victoria Infirmary, Newcastle upon Tyne Hospitals NHS Foundation Trust, Newcastle upon Tyne, UK

[3]Institute of Health and Society, Newcastle University, Newcastle upon Tyne, UK

[4]NIHR Newcastle In Vitro Diagnostics Co-operative, Newcastle upon Tyne Hospitals NHS Foundation Trust, Newcastle upon Tyne, UK

[5]Cardiology Department, Medical School, University of Belgrade, Clinical Centre Serbia and Serbian Academy of Science and Arts, Belgrade, Serbia

[6]Darlington Memorial Hospital, County Durham and Darlington NHS Foundation Trust & School of Medicine, Pharmacy and Health, Durham University, Durham, UK

[7]Primary Care Unit, Department of Public Health and Primary Care, University of Cambridge, Cambridge, UK

[8]Nuffield Department of Primary Health Care Sciences, University of Oxford, Oxford, UK

[9]RCUK Centre for Ageing and Vitality, Newcastle University, Newcastle Upon Tyne, UK

**Correction notice** This article has been corrected since it was published online. The affiliations of authors have been updated.

**Acknowledgements** The authors would like to thank all healthcare professionals who gave their time to being interviewed and participating in this study.

**Contributors** DGJ, SC, NO, GM and SG: study concept and design, drafting of the manuscript; SC and NO: acquisition of data, administrative, technical or material support; SC, NO, GM and SG: data analysis and interpretation of data; TM, HB, LV, GAM, PMS, AF, CD, JM and RH: critical revision of the manuscript for important intellectual content. DGJ: study supervision. All authors critically reviewed the manuscript and approved the final version. DGJ acts as the guarantor and takes responsibility for the content of the manuscript, including the data and analysis.

**Funding** This study was funded by the UK Medical Research Council Confidence in Concept Scheme grant to DGJ (grant no. BH161161). SC and NO are supported by the European Horizon 2020 research and innovation programme under grant agreement no 777204. SG is supported by the NIHR MEDTEch In Vitro Diagnostics. DGJ is supported by the UK Research Councils' Newcastle Centre for Ageing and Vitality (grant no. L016354).

**Disclaimer** The views expressed are those of the authors and not necessarily those of the Medical Research Council. The funders had no role in study design or in data collection, analysis or interpretation.

**Competing interests** None declared.

**Patient consent for publication** Not required.

**Ethics approval** The study protocol (number 16/NE/0287) was approved by the National Health Service, Health Research Authority (North East – Tyne & Wear South Research Ethics Committee). All procedures performed in the study were in accordance with the Declaration of Helsinki.

**Provenance and peer review** Not commissioned; externally peer reviewed.

**Data sharing statement** The relevant anonymised patient-level data are available on reasonable request from the authors.

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
