## [Reviewer comments · BMJ Open]

ARTICLE DETAILS

TITLE (PROVISIONAL)	Opportunities and challenges of a novel cardiac output response to stress (CORS) test to enhance diagnosis of heart failure in primary care: a qualitative study
AUTHORS	Charman, Sarah; Okwose, Nduka; Maniatopoulos, Gregory; Graziadio, Sara; Metzler, Tamara; Banks, Helen; Vale, Luke; MacGowan, Guy; Seferović, Petar; Fuat, Ahmet; Deaton, Christi; Mant, Jonathan; Hobbs, Richard; Jakovljevic, Djordje

VERSION 1 - REVIEW

REVIEWER	Witte, Klaus University of Leeds
REVIEW RETURNED	15-Dec-2018

GENERAL COMMENTS	This is a well written qualitative piece assessing the potential utility and uptake of a novel cardiac output device to help stratify patients presenting to primary care with signs, symptoms of heart failure. I am not a qualitative researcher, but in my limited experience the methodology is sound, and the results are valid with the appreciated limitation of the small sample size of discussants. The manuscript raises the important points around access to secondary care for the large number of patients with features suggestive of heart failure and how services could be adapted to cope with this. It's well written and readable. It's relevant to current practice and provides a first impression of one potential solution.
---

REVIEWER	Karin Hellström Ångerud Department of Nursing, Umeå University, Sweden
REVIEW RETURNED	28-Dec-2018

GENERAL COMMENTS	In this qualitative study the authors aimed to determine primary and secondary care healthcare professional's perceptions of the current clinical care diagnostic pathway for heart failure and the potential implementation a novel Cardiac Output Response to Stress (CORS) test developed to improve diagnosis of heart failure. Fourteen healthcare professionals were interviewed with semi-structured in-depth interviews. The text were analysed thematically using an inductive approach, resulting in six broad themes.
---

	Comments:  - This study include a twofold aim, and the rational for the first part is not clear to me. - Some information about the participants is missing (e.g. working experience, gender, age). It is also not clear if the participants have worked with the new diagnostic test. - A more detailed description about how the thematic analysis was performed is needed. The steps in the analysis process are not described. - The results correspond mainly to the second part of the aim (how health care professionals perceive a new diagnostic test). In figure 2, it is not clear what is results from interviews and what is from NICE guidance.
--	--

REVIEWER	Sara Schröder Institute of Medical Sociology, Medical Faculty, Martin Luther University Halle-Wittenberg, Halle, Germany
REVIEW RETURNED	30-Dec-2018

GENERAL COMMENTS	Thank you for the opportunity to review this manuscript. The authors explore the opportunities and challenges of implementing a novel test of diagnosing HF in GB. In general, this is an interesting study, but I have major concerns regarding the objective, results and discussion, that need to be edited before a decision of acceptance can be done. First, the objective is not consistent throughout the manuscript whether it is “opportunities and challenges of the CORS test” or “perceptions of the potential implementation of CORS” or “evaluation of the current clinical pathway of HF”. This reflects also my second concern, as the study mixes up two objectives as Theme 1 is the rationale for developing a novel test for diagnosing HF and Themes 2-6 reflect the opportunities and challenges of the implementation. It seems that the “clinical need in the current pathway” was rather a presupposition because one topic of the interviews was the implementation of a novel test. Can you give the rationale for describing the current clinical pathway in the results (and not e.g. in the introduction) as the relation to the objective and other results is missing and its only outlined in one sentence and one figure? This becomes also clear as the discussion starts with the key point of the rationale for implementing a novel diagnostic tool on Page 14, line 17-21 and is not associated with the research aim. Third, the results and mayor themes should be more specific: The six broad themes are really broad and the reader can hardly understand the results when reading the abstract as they do not outline the opportunities and challenges clearly. Can you describe them more detailed in the abstract and first section of the results with regard to the aim of the study? The Term of Theme 3 is misleading as it rather reflects “patient exclusion and suitability” than “usability and acceptability”. The explanation of Theme 4 is confusing and the context to the term “costs and resources” is not worked out quite plausible. In
--

	addition, Theme 4, 5 and 6 seem to reflect the same issue. The content should be differentiated more clearly. Fourth, the study remains very specific in the context of the CORS test, but would be more appropriate for international readers and implementation research when discussed in the overall context of opportunities and challenged of implementing novel diagnostic tool in primary care. In the discussion the clinical need and implementation of the CORS test is addressed but not the results of opportunities and challenges. Are there other studies, which have found similar opportunities and challenged of implementing novel diagnostic tool in primary care? Lastly, I have some minor concerns:  1. Generally, introduce abbreviations when they are used the first time and use only the abbreviation in the following (e.g. page 5, line 37; Page 6, line 14). Abstract:  2. The conclusion rather repeats two of the results themes. Introduction:  3. The sentence on Page 4, line 30-37 is hardly understandable. 4. Page 4, line 46: please explain “creating unnecessary burden and costs to secondary care” more detailed. 5. Page 5, line 5-10: please specify more detailed the rationale for the study: why did you assessed it important to understand how healthcare professionals perceive the CORS test. Methods:  6. Page 6, line 10-19: Could you give examples of the questions asked? 7. Did you reach data saturation?? Can you give a rationale for the number of 14 participants? 8. Page 6, line 40-53: data analysis is described weakly. Can you explain why transcripts were grouped and subdivided? Results:  9. Do the sentences on page 13, line 57-60 belong to the theme 6 or are they an overall statement. Then they should be moved to the beginning of the results section, as the context is misleading when they are placed here.
--	--

VERSION 1 – AUTHOR RESPONSE

REVIEWER #1:

Comment 1: This is a well written qualitative piece assessing the potential utility and uptake of a novel cardiac output device to help stratify patients presenting to primary care with signs, symptoms of heart failure.

Response 1: The Authors appreciate this comment.

Comment 2: The methodology is sound, and the results are valid with the appreciated limitation of the small sample size of discussants.

Response 2: The authors appreciate this comment.

Comment 3: The manuscript raises the important points around access to secondary care for the large number of patients with features suggestive of heart failure and how services could be adapted to cope with this.

Response 3: Thank you for the comment.

Comment 4: It's well written and readable. It's relevant to current practice and provides a first impression of one potential solution.

Response 4: Thanks for the comment. This comment underlies the primary aim of the present manuscript.

REVIEWER #2:

Comment 1: This study include a twofold aim, and the rational for the first part is not clear to me.

Response 1: Thank you for the comment. The aims/objective of the study has been revised to ensure consistency. This could be found in abstract 'Objective' and also in page 6, lines 10-12. The first part of the interviews which focused on the clinical care pathway for heart failure patients was done in order to find out what part of the pathway the CORS test could potentially fit into.

Comment 2: Some information about the participants is missing (e.g. working experience, gender, age). It is also not clear if the participants have worked with the new diagnostic test.

Response 2: The Participant information have now been updated in table 1 (page 21) to include working experience and gender. The study team did not ask for age of interviewees as we thought working experience in the field of heart failure was a better descriptor with respect to the present study's goals. The participants have not previously worked with the new test. They were only given a demonstration of how the test works and this reflected under Methods section 'The CORS test' and in figure 2.

Comment 3: A more detailed description about how the thematic analysis was performed is needed. The steps in the analysis process are not described.

Response 3: The detailed explanation on how thematic analysis was performed have now been added to the method section under 'data analysis' (page 7/8 lines 1-12).

Comment 4: The results correspond mainly to the second part of the aim (how health care professionals perceive a new diagnostic test). In figure 1, it is not clear what is results from interviews and what is from NICE guidance.

Response 4: Thank you for the comment. Figure 1 describes the clinical pathway, which GP practices follow in the treatment of Heart failure. This is in line with the NICE guideline and not a deviation. We have also revised all themes so they are coherent with the objectives of the study.

REVIEWER #3:

Comment 1: The authors explore the opportunities and challenges of implementing a novel test of diagnosing HF in GB. In general, this is an interesting study, but I have major concerns regarding the objective, results and discussion, that need to be edited before a decision of acceptance can be done.

Response 1: Thank you for your comment. We appreciate there are inconsistencies and we have modified the objectives to ensure consistency and clarity throughout the manuscript.

Comment 2: First, the objective is not consistent throughout the manuscript whether it is “opportunities and challenges of the CORS test” or “perceptions of the potential implementation of CORS” or “evaluation of the current clinical pathway of HF”.

Response 2: Modifications have been made to ensure clarity of the manuscript. First, we seek to establish potential role of the CORS test in the clinical pathway for heart failure management and then we further discuss the opportunities and challenges for implementation. This has been modified and reflects in the abstract objective and also introduction page 6 lines 10-12.

Comment 3: This reflects also my second concern, as the study mixes up two objectives as Theme 1 is the rationale for developing a novel test for diagnosing HF and Themes 2-6 reflect the opportunities and challenges of the implementation. It seems that the “clinical need in the current pathway” was rather a presupposition because one topic of the interviews was the implementation of a novel test. Can you give the rationale for describing the current clinical pathway in the results (and not e.g. in the introduction) as the relation to the objective and other results is missing and its only outlined in one sentence and one figure? This becomes also clear as the discussion starts with the key point of the rationale for implementing a novel diagnostic tool on Page 14, line 17-21 and is not associated with the research aim.

Response 3: This comment is well thought out and we appreciate and acknowledge the inconsistency. The introduction have now been revised and a brief note added on the clinical care pathway for heart failure (page 5, lines 8-16) and the potential role for an add-on test to improve diagnosis.

Comment 4: Third, the results and mayor themes should be more specific: The six broad themes are really broad and the reader can hardly understand the results when reading the abstract as they do not outline the opportunities and challenges clearly. Can you describe them more detailed in the abstract and first section of the results with regard to the aim of the study? The Term of Theme 3 is misleading as it rather reflects “patient exclusion and suitability” than “usability and acceptability”. The explanation of Theme 4 is confusing and the context to the term “costs and resources” is not worked out quite plausible. In addition, Theme 4, 5 and 6 seem to reflect the same issue. The content should be differentiated more clearly.

Response 4: As per your suggestion, all themes have been revised and rephrased (Themes 1-4 pages 8, 10,11and 12) to make for better meaning and to reflect the description given. The abstract results section have also been revised. Themes 4-6 have been compressed into one theme (theme 4 page 12 line 8) for clarity and better understanding.

Comment 5: Fourth, the study remains very specific in the context of the CORS test, but would be more appropriate for international readers and implementation research when discussed in the overall context of opportunities and challenged of implementing novel diagnostic tool in primary care. In the discussion the clinical need and implementation of the CORS test is addressed but not the results of opportunities and challenges. Are there other studies, which have found similar opportunities and challenged of implementing novel diagnostic tool in primary care?

Response 5: The ‘discussion’ section (page 14) has been drafted to reflect the broader discussion of opportunities and challenges of implementing novel diagnostic tool in primary care. However, the present study presents a unique challenge, as this is the first study of its kind in cardiac care and other studies have only been reported to give a sense of the barriers (page 15, lines 2-8) encountered in the implementation of other forms of diagnostic tests.

REVIEWER #3 Minor Concerns:

Comment 1: Generally, introduce abbreviations when they are used the first time and use only the abbreviation in the following (e.g. page 5, line 37; Page 6, line 14).

Response 1: Thank you, the mistakes have been duly corrected.

Comment 2: Abstract: The conclusion rather repeats two of the results themes.

Response 2: This has been rephrased in the abstract section.

Comment 3: Introduction: The sentence on Page 4, line 30-37 is hardly understandable.

Response 3: This has been revised to ensure better understanding (Page 5, lines 16-23).

Comment 4: Page 4, line 46: please explain "creating unnecessary burden and costs to secondary care" more detailed.

Response 4: The clarity has not been provided (page 5, lines 16-23).

Comment 5: Page 5, line 5-10: please specify more detailed the rationale for the study: why did you assessed it important to understand how healthcare professionals perceive the CORS test.

Response 5: A short paragraph has been added to explain this (Page 6, lines 3-12).

Comment 6: Methods: Page 6, line 10-19: Could you give examples of the questions asked?

Response 6: A sample topic guide and interview questions has been added to the Appendix.

Comment 7: Did you reach data saturation?? Can you give a rationale for the number of 14 participants?

Response 7: Yes we reached data saturation and no further themes directly related to our objectives were emergent. We wanted to receive responses from a cross section of persons involved directly or indirectly in Heart failure patient care and possible implementation of the CORS test and the study team thought interviewing at least two people from this spectrum would be appropriate.

Comment 8: Page 6, line 40-53: data analysis is described weakly. Can you explain why transcripts were grouped and subdivided?

Response 8: We have revised our data analysis section (page 7/8 lines 1-14) and buttressed our use of codes which were later compressed to form themes. Transcripts were initially grouped according to primary and secondary care interviews and data analysed according to themes flowing from that but on further analysis there were overlapping codes and themes thus the final set of themes. The initial statement about transcripts been grouped and subdivided is an error part.

Comment 9: Do the sentences on page 13, line 57-60 belong to the theme 6 or are they an overall statement. Then they should be moved to the beginning of the results section, as the context is misleading when they are placed here.

Response 9: Thank you for your observation. Indeed, these are general statements and they have been deleted, as it is only a repetition of the overall idea projected in the themes.

VERSION 2 – REVIEW

REVIEWER	Sara Schröder Institute of Medical Sociology, Martin Luther University Halle-Wittenberg, Halle (Saale), Germany
REVIEW RETURNED	03-Mar-2019

GENERAL COMMENTS	Thanks for the opportunity to review this study again. The manuscript was revised entirely, but some issues haven't been addressed carefully. Therefore, I still have some minor concerns, which are drawn up below. Many questions of the interview schedule are closed yes/no-questions, which are not suitable for in-depth interviews, like claimed in page 5. Can you please portray the in-depth key questions with regard to the two research questions directly in the manuscript. In your response you write that "The initial statement about transcripts been grouped and subdivided is an error part." But this error information is still in the revised manuscript on page 6. In the results you display the "four themes relating to opportunities and challenges surrounding the implementation". The first theme is the "role of the test in the current pathway" which is likewise the objective of the study. The objective should not be the result.
--

VERSION 2 – AUTHOR RESPONSE

REVIEWERS' COMMENTS AND AUTHORS' RESPONSES

REVIEWER #1:

Comment 1: The manuscript was revised entirely, but some issues haven't been addressed carefully. Therefore, I still have some minor concerns, which are drawn up below.

Response 1: Thank you. We have now followed your further suggestions and revised the manuscript to ensure our objectives are clearly expressed throughout the manuscript.

Comment 2: Many questions of the interview schedule are closed yes/no-questions, which are not suitable for in-depth interviews, like claimed in page 5. Can you please portray the in-depth key questions with regard to the two research questions directly in the manuscript.

Response 2: Thank you for this insightful comment. As noted in manuscript, the interviews lasted on average one hour and we acknowledge that some of the interview questions were closed. However, prompts were used to further generate responses to the questions asked. We have now included the prompts (red fonts) in the interview schedule and have highlighted key questions in yellow (please find the HCP Interview Schedule_v1_R2 document enclosed / attached). As you also suggest, we have now indicated additional key questions that directly relate to our objectives in the manuscript on Page 6 lines 7-16. We hope this satisfies but please let us know if there are any further suggestions.

Comment 3: In your response you write that "The initial statement about transcripts been grouped and subdivided is an error part." But this error information is still in the revised manuscript on page 6.

Response 3: Thank you for this observation. This has now been deleted.

Comment 4: In the results, you display the "four themes relating to opportunities and challenges surrounding the implementation". The first theme is the "role of the test in the current pathway" which is likewise the objective of the study. The objective should not be the result.

Response 4: Thank you. We have now revised the first theme for clarity purposes. New theme can be found under results page 7, lines 13 and 17.